# RER2023: the landslide inventory dataset of the May 2023 Emilia-Romagna meteorological event

Matteo Berti [1], Marco Pizziolo [2], Michele Scaroni [2], Mauro Generali [2], Vincenzo Critelli [3], Marco Mulas[3], Melissa Tondo [3], Francesco Lelli [3], Cecilia Fabbiani [3], Francesco Ronchetti [3], Giuseppe Ciccarese [1], Nicola Dal Seno [1], Elena Ioriatti [1], Rodolfo Rani [1], Alessandro Zuccarini [1], Tommaso Simonelli [4], Alessandro Corsini [3].

[1]Department of Biological, Geological, and Environmental Sciences, University of Bologna, Bologna, Italy

[2]Regione Emilia-Romagna, Area Geologia, Suoli e Sismica, Bologna, Italy

[3]Department of Chemical and Geological Sciences, University of Modena and Reggio-Emilia, Modena, Italy

[4]Autorità di Bacino Distrettuale del Fiume Po, Parma, Italy

*Correspondence to:* Alessandro Corsini (alessandro.corsini@unimore.it)

**Abstract.** Landslide inventories play a vital role in assessing susceptibility, hazards, and risks, and are essential for developing resilience strategies in mountainous areas. This importance is amplified in the context of climate change, as existing inventories might not adequately reflect changing stability conditions. In May 2023, the Emilia-Romagna region

of Italy was hit by two major rainfall events, leading to widespread flooding and the triggering of thousands of landslides. Predominantly, these were shallow debris slides and debris flows, occurring on slopes previously deemed stable based on historical data with no prior landslides recorded. Our team supported the Civil Protection Agency through field surveys and mapping efforts to pinpoint and record these landslides, prioritizing areas critical to immediate public safety and focusing on thorough mapping for future recovery planning. The outcome is a detailed map of all landslides induced by

these events, manually identified using high-resolution aerial photography (0.2 m pixel resolution, RGB+NIR four bands) and categorized with the help of a 3D viewer. This comprehensive landslide inventory, comprising 80997 landslide polygons, has been made openly accessible to the scientific community.

## 1 Introduction

Landslide inventories are crucial for susceptibility, hazard, and risk assessments and management (Soaters & Van Westen,

1996; Fell et al., 2008; Galli et al., 2008; Corominas et al., 2014). In Europe, landslides inventories are compiled on a national to a regional basis (Van Den Eeckhaut and Hervás, 2012) and can be supported by advanced landslides recognition and monitoring techniques (Guzzetti et al., 2012; Jaboyedoff et al., 2012; Amatya, 2021; Catani, 2021; Bhuyan et al., 2023). Landslides inventories should be as complete and spatially accurate as possible and, also, they should consistently distinguish and classify different landslides types. These factors are important for improving

frequency-area analyses (Malamud et al., 2004) and for obtaining reliable statistically-based landslides susceptibility maps, thanks to complete input data (Steger et al., 2017; Gaidzik et al., 2021) and disjunct analysis of landslides types (Zêzere, 2002).

Generally, inventories of large-scale landslides are quite complete, since their geomorphic features that remain evident long after their occurrence and they can have slow movements detectable by remote sensing (Bertolini et al., 2017; Rosi

et al., 2018; Luetzenburg et al., 2022; Ardizzone et al., 2023). On the contrary, regional or national inventories might not include a complete record of past shallow rainfall induced landslides, unless they have been mapped soon after occurrence, i.e. before becoming hardly recognizable due to vegetation growth, rill erosion, or land cultivation (Guzzetti et al., 2004; Crozier, 2005; Cardinali et al., 2006; Zieher et al., 2016; Hao et al., 2020, Santangelo et al., 2023; Pittau et al., 2024). Therefore, it is crucial that existing national and regional landslide inventories are updated following each significant rainfall event to gather data that are not only essential but also enhance the systematic application of landslide susceptibility maps in land-use planning (Fell et al., 2008). This is a challenging and important task, as the incidence of shallow rainfall induced landslides is likely to increase in Europe due to climate change (Gariano and Guzzetti, 2016; Handwerger et al., 2022; Auflič et al., 2023).

In the Emilia-Romagna Region (Northern Italy), land-use planning and land-use restrictions are based on an inventory map of landslides at 1:10.000 scale (Bertolini et al., 2005) and on a catalogue of thousands of records referring to the activation or reactivation of landslides in the past (Piacentini et al., 2018). These documents are updated after every occurrence or reactivation of large-scale landslides and after multiple occurrences of rainfall-induced shallow landslides. In recent years, updates have been necessary to include hundreds of debris flows triggered during the rainstorm events that hit Parma province in September 2014 (Corsini et al., 2017) and Piacenza province in October 2015 (Scorpio et al., 2018; Ciccarese et al., 2020). In May 2023, the entire southern sector of Emilia-Romagna (from Rimini to Reggio Emilia provinces) has been hit by two consecutive exceptional rainfall events that triggered thousands of first-failure landslides. Ferrario and Livio (2023) provided an initial screening of these landslides through visual inspection of Planet satellite images (3 m resolution) and limited field surveys, while Notti et al. (2024) utilized an unsupervised identification method on Sentinel 2 images (10 m resolution). Both inventories were quickly compiled soon after the emergency, utilizing satellite imagery with resolutions that are relatively low considering the small size of the landslides from May 2023. Indeed, many of these landslides were so small that they were challenging to detect even at a 3 m resolution. Additionally, landslides were not classified or categorized by type of movement or material. Consequently, these two datasets were not intended to achieve, nor did they achieve, the level of completeness, consistency, and accuracy required for updating the official landslide inventory.

In this paper, we present the landslide inventory dataset of the May 2023 Emilia-Romagna events which has been designated as reference map by the Emilia-Romagna Region and the Po River Authority for the "Special Plan for interventions against situations of hydrogeological instability" (approved in preliminary version in April 2024), to aid the Commission for Reconstruction in implementing the recovery phase. This spatial dataset is based on expert-based identification, mapping and classification of landslides on high-resolution aerial images taken shortly after the second event (0.2 m resolution, RGB and near-infrared). Particular attention has been given to the consistency of landslides type classification, which required development and application of an algorithm for data harmonization across areas surveyed by different operators.

## 2 The May 2023 Emilia-Romagna event

The Emilia-Romagna region is located in northern Italy and stretches from the Apennine Mountains to the Po River Valley and eastward to the Adriatic Sea (Fig. 1a). It is one of Italy's most economically prosperous regions, with a strong industrial base in automotive, machinery, food processing, and ceramics. The Po River Valley plays a vital role in agriculture, while the eastern coastline is a hub for both domestic and international tourism. By contrast, the Apennine Mountains present a more subdued economic landscape. Economic activity and population in these mountains have

declined since the 1960s, and now focus on agrotourism, ecotourism, and niche markets. Regional initiatives are underway
to foster economic growth in these mountainous areas.

In May 2023, the Emilia-Romagna region was struck by two exceptional rainfall events. The first, from May 1-3, delivered
approximately 200 mm of rain over a span of 48 hours. Only two weeks later, on May 16-17, a second event matched this
intensity, with rainfall totals reaching 200-250 mm within another 48-hour window. The recurrence interval for a single
two-day event was estimated to exceed 100-300 years, but the combined effect of these two closely timed events far
surpassed 500 years (Brath et al., 2023). Both events impacted roughly the same area in the eastern part of the region
(Fig. 1).

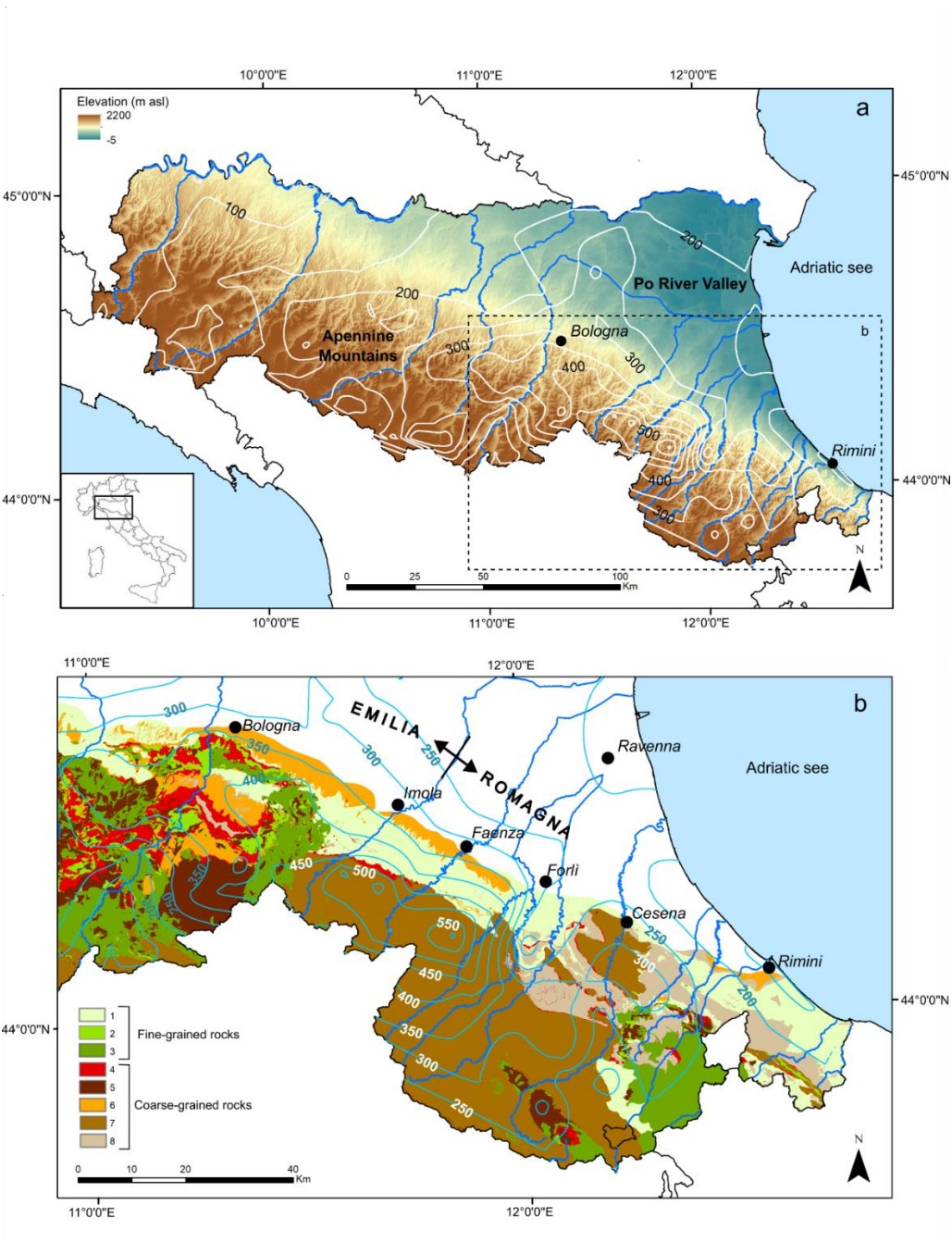

**Fig. 1. a) Overview map of the Emilia-Romagna region (Italy), illustrating elevation and cumulative rainfall isohyets from May 1-17, 2023. b) Detailed view of the area most impacted by the event, featuring the geological units referenced in Table 1.**

These rainfalls led to extensive flooding across the Po plain and triggered thousands of landslides in the Apennines. The total damages have been estimated to surpass 9 billion euros, affecting roads, railways, buildings, and cultural heritage sites, along with the destruction of bridges, power facilities, and communication lines. Additionally, agricultural fields, farming operations, and cultivated slopes saw significant disruption over an area of about 1000 km². Fifteen people lost their lives due to the flooding and two due to landslides.

The Emilia-Romagna region and the Italian Government promptly responded to the event mobilizing all necessary resources. The primary focus was on the Po plain area, which is densely populated and houses the majority of industrial and agricultural activities. Consequently, the severe issues caused by landslides in the mountainous regions were initially overlooked. Over time, the significance of these issues became apparent, but even a year after the disaster the situation remains critical. The impact of landslides in the Apennines has been especially severe due to the local economy's

vulnerability, the extensive damages to infrastructure, and the significant land loss, all of which have slowed and complicated the recovery process.

We assisted local and national agencies and working groups in addressing the problems caused by landslides. The initial two weeks following the event were primarily focused on field surveys and rapidly assessment of the most critical situations that demanded immediate actions to ensure public safety. Subsequently, our efforts shifted towards landslide

mapping. In a first stage, it was crucial to identify the roads and buildings affected by landslides to coordinate emergency interventions and perform an initial damage assessment. Afterward, we completed the landslide inventory to develop a comprehensive map detailing all landslides triggered by the event across the area. This map has been officially designated as the landslide map for the May 2023 event by the Po River Authority and the Emilia-Romagna region, and it is currently being used by the Commission for Reconstruction for implementing the recovery phase.

**3 Methods**

This section describes the methodology used to develop the landslide inventory for the May 2023 event. It covers the classification of lithological units, the identification and mapping of landslides, their classification, and the quality control and data harmonization procedures implemented.

**3.1 Lithological units classification**

In the study area, bedrock geology significantly influences the morphology of the slopes, the mechanical properties of the weathered soil layer, the vegetation cover and, consequently the proneness to slope instability. As a matter of fact, these factors played a crucial role in the behavior of the slopes during the May 2023 event, controlling the type and density of landslides. Consequently, bedrock geology is an essential base layer of our landslide inventory.

The geological map of the Emilia-Romagna region, created by the regional Geological Survey at a 1:10.000 scale (AGSS-

RER, 1986), includes more than 600 geological formations. These formations are distinguished by unique features that signify variations in depositional environments, composition, or geological age. The variety and detail of these formations illustrate the region's complex geological history and the precision employed in the map's creation.

For our inventory, we categorized all geological formations into eight distinct units, as depicted in Fig. 1b and detailed in Table 1. These units were delineated by merging lithological characteristics with their respective structural domains,

recognizing that the same rock type can display varying structural and mechanical properties depending on its location within the orogenic sequence. For example, flysch rocks within the Ligurian domain (unit 5) are generally more fractured,

less resistant, and prone to deep-seated landslides compared to those in the Tuscan-Umbrian domain (unit 7), due to the extensive tectonic stress they endured in the accretionary wedge.

| Unit ID | Lithology | Domain | Structural position | Geological Age |
|---|---|---|---|---|
| 1 | Clays, silty clays, and marly clays | Padano-Adriatic | Outer Foredeep | Pliocene to Pleistocene |
| 2 | Marls and marly clays | Epiligurian | Wedge-top basins | Oligocene to Miocene |
| 3 | Clay shales, clay breccias, tectonized clays, olistostromes | Ligurian | Accretionary wedge | Cretaceous to Eocene |
| 4 | Massive rocks: basalts, serpentines, limestones, arenites | Ligurian, Epiligurian | Accretionary wedge Wedge-top basins | Cretaceous to Miocene |
| 5 | Flysch rocks made of rhythmic alternations of sandstones, limestones, pelites, and shales | Ligurian, Epiligurian | Accretionary wedge Wedge-top basins | Cretaceous to Eocene |
| 6 | Weakly cemented sandstones and conglomerates | Padano-Adriatic | Outer Foredeep | Pliocene to Pleistocene |
| 7 | Flysch rocks made of rhythmic alternations of sandstones and pelites | Tuscan-Umbrian | Inner Foredeep | Miocene |
| 8 | Weakly cemented sandstones with interbedded pelitic layers | Padano-Adriatic | Outer Foredeep | Pliocene to Pleistocene |

**Table 1. Classification of the geological formations in the Emilia-Romagna region into eight units, based on their lithological composition and geological structural domains. Units 1 to 3 consist mainly of fine-grained rocks, while units 4 to 8 are primarily composed of coarse-grained rocks.**

The eight units identified were further divided into two broad categories: fine-grained rock masses (units 1 to 3) and coarse-grained rocks (units 4 to 8). This categorization aids in the preliminary differentiation of the types of weathered soil covers these rocks produce, which experienced widespread landslides during May 2023. Coarse-grained rocks typically produce granular soils composed of sand, gravel, and cobbles, with smaller amounts of silt and clay, aligning with the "debris" category in the Cruden and Varnes (1996) classification. In contrast, fine-grained rocks lead to the formation of fine soils predominantly made up of silt and clay, fitting the "earth" classification. These two categories, "debris" and "earth," are utilized to classify landslides that occurred on soil-covered slopes.

### 3.2 Landslide identification and mapping

Landslides identification and mapping was conducted by means of photo-interpretation of high-resolution aerial images. These images were captured using a Leica DMC III sensor aboard a Cessna 402C aircraft, flying at approximately 4700 meters above sea level. The images, taken shortly after the second rainfall on May 23, 2023, have a 0.2 m resolution and include four bands: RGB and near-infrared.

The mapping process was organized as follows. The total area was segmented based on the administrative boundaries of the municipalities. These sections were then distributed among three institutions: the University of Bologna, the University of Modena-Reggio Emilia, and the Geological Survey of the Emilia-Romagna Region. Each institution assigned four mappers, with a total of twelve individuals involved in the effort. Landslide detection was conducted in GIS environment by comparing pre- and post-event images with an on-screen zoom of approximately 1:1000 (Fig. 2A and B). Once a landslide was spotted, further inspection was conducted using NIR and NDVI images (Fig. 2C and D) and supplemented by a 3D viewer featuring high-resolution images overlaid on a 30 m DEM (Copernicus GLO-30, ESA, 2024; Fig. 2E). Viewing the slope from different angles enhanced the delineation of the affected area and the interpretation of the type of movement. Following this analysis, each landslide was classified into the specified classes described in the next section. The digital mapping of the landslide polygon was then executed at scales ranging from 1:800 to 1:200,

depending on the landslide's size, ensuring precise tracing of the affected perimeter. In this final stage, the regional topographic map at 1:10,000 scale was utilized to further verify alignment with the existing topography (Fig. 2F). While the delineation of the landslides was carried out at a large scale for precise mapping of the boundaries, the final inventory is designed to be appropriate for consultation at a scale of 1:2000.

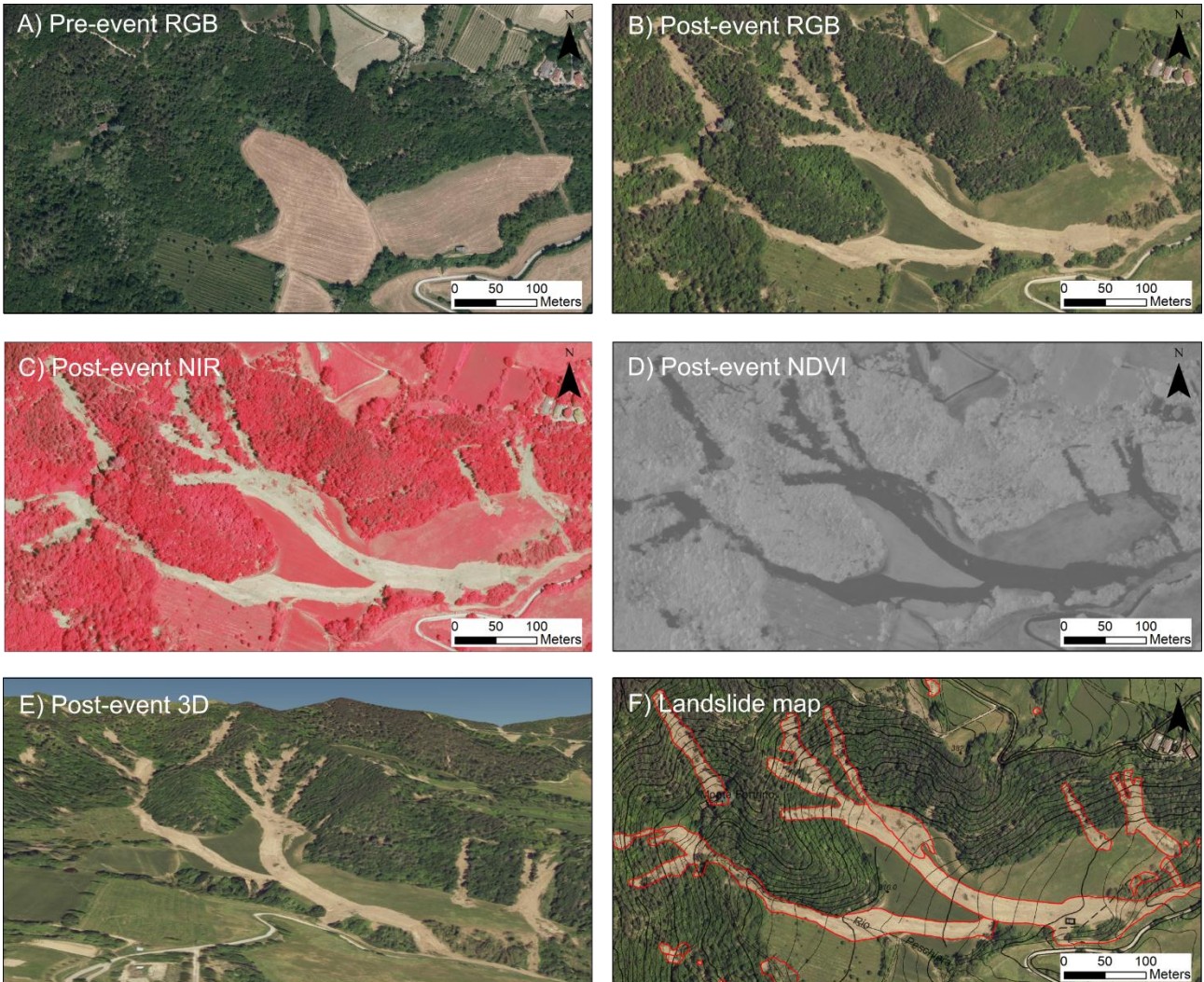

**Fig. 2. Example of manually identifying and mapping landslides. A) Pre-event image (AGEA aerial photos, April-July 2020, 0.2 m resolution); B) Post-event image (May 23, 2023, 0.2 m resolution); C) Near Infrared (NIR) images derived from post-event images; D) Normalized Difference Vegetation Index (NDVI) images derived from post-event images (dark colors signify absence of vegetation); E) 3D visualization of post-event images; F) Landslide map manually created using A to E and aligned with the pre-event topographic map at a 1:10,000 scale.**

Overall, the process of identifying landslides was fairly objective. Each landslide cleared vegetation, uncovered distinct patches of bare soil or bedrock, and led to deposits of loose material. The post-event images, taken only 10 days after the second rainfall, clearly displayed these geomorphological markers, eliminating any ambiguity in recognizing the landslides from the event. Additionally, most of the landslides were new occurrences, not present in the pre-event images; even in case of reactivations, it was straightforward to identify the newly affected areas. This gives us confidence that our dataset exclusively contains landslides from the May 2023 event. It is important to emphasize the distinct nature of this dataset. The existing landslide map of Emilia-Romagna Region (https://geoportale.regione.emilia-romagna.it/) encompasses all landslides identified in the region via photo interpretation, historical data, and field surveys. This map

provides a comprehensive overview of landslide activity across a broad geological timeframe, accounting for various climatic, seismic, and morphological conditions. In contrast, our dataset specifically captures the landslide response to a single critical meteorological event. Additionally, the criteria used to distinguish different types of landslides are specifically designed for this scenario, as explained in the subsequent section (see 3.3).

While identifying the landslides was straightforward, defining their exact boundaries was more subjective. Many landslides became fluidized upon failure, with the distal debris spreading among trees without removing vegetation, thus complicating the mapping of the deposit (Fig. 3a). Moreover, several slopes experienced complete removal of soil cover by adjoining shallow failures, blurring the distinction between individual slides (Fig. 3b). In these cases, we chose to interpret the landslide boundaries rather than just tracing the visible debris edges. For fluidized slides, the polygons were adjusted by connecting visible debris patches to include areas obscured by vegetation. For coalescent slides, we attempted to map each individual slide by identifying distinctive arcuate shapes along the detachment scarps that signified separate failures. Although this approach introduced some subjectivity into the manual mapping process, it was necessary to create a dataset suitable for analyzing the morphometric features of the landslides.

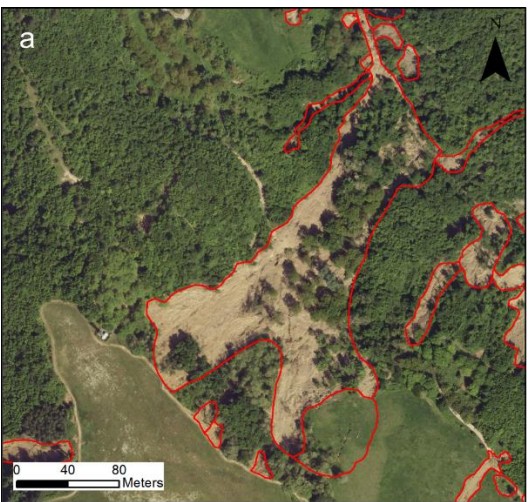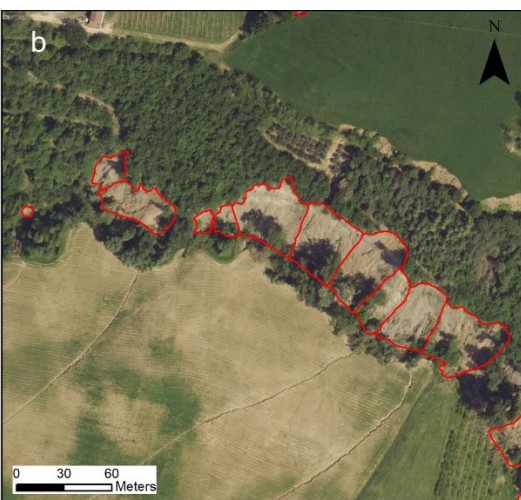

**Fig. 3. Examples of uncertain landslide boundary delineations. a) Debris slides interspersed among trees without clearing the vegetation; b) multiple coalescent debris slides segmented into individual slides.**

The manual mapping process was demanding and labor-intensive. Initially, we focused on mapping landslides in the areas most severely impacted by the event, particularly around roads and urban centers. This priority was set to align with the Civil Protection Agency's needs to identify damages during the emergency response. This initial phase of mapping, which produced several "damage maps" for the affected municipalities, spanned the first two months following the disaster. Subsequently, the landslide inventory was expanded to cover the entire area over the following months. The complete process took approximately six months to finish.

### 3.3 Landslides classification

Right from the start of our work, the classification of landslides triggered by the event was identified as a crucial task. We utilized the well-known Cruden and Varnes (1996) classification system, which categorizes landslides based on two primary criteria: the type of movement and the type of material. Most of the landslides from the May 2023 event were categorized either as debris slides, where mixed granular material moves along a plane of weakness as a relatively coherent mass, or as debris flows, where the material moves in a fluid-like manner over greater distances. However, we

encountered challenges in further distinguishing between various types of debris slides and debris flows, a differentiation not addressed by the standard classification.

Figures 4 and 5 illustrate the problem. According to the Cruden and Varnes (1996) classification, all the six landslides depicted in the Fig. 4 can be classified as debris flows. Yet, clear differences are apparent between the upper (a1-3) and the lower three (b1-3). The latter are typical debris flows that start on a steep slope and stop as the slope decreases; the

former, while starting similarly on steep slopes, demonstrate extensive propagation and complete fluidization of the deposit as they travel much further. None of these cases involve a well-defined channel, thus the classification proposed by Hungr et al. (2014) that distinguished channelized debris flows from unchannelized debris avalanches does not apply here. A similar challenge presents with debris slides (Fig. 5). Current classifications fail to distinguish between slides of different degrees of mobility, a distinction that is clearly visible in the field. Some slides exhibited in fact high mobility,

completely clearing the vegetation (upper pictures a1-3), while others show low mobility, indicated by minimal vegetation damage (lower pictures b1-3). Understanding the conditions that lead to these diverse behaviors is crucial for hazard assessment and necessitates differentiating these phenomena.

An additional classification challenge involves rock-block slides. These landslides impacted the homoclinal slopes of the Marnoso Arenacea Formation (unit 7 in Fig. 1b) and manifested as massive, translational rock-slab slides along bedding

planes. While classifying these landslides poses no issues, it was necessary to distinguish between rock slides based on their degree of evolution. Some experienced movements ranging from several meters to tens of meters, signaling paroxysmal failures (Fig. 6 a1-a3), whereas others shifted merely a few centimeters, indicative of incipient, undeveloped failures (Fig. 6 b1-b3). The latter represent highly dangerous zones prone to potential collapse and thus required special attention.

**DF1)** Debris flows with runout extending on gentle unforested slopes

**DF2)** Debris flows with runout limited on steep forested slopes

**Fig. 4. Representative images of the two distinct types of debris flows caused by the May 2023 event.**

**DS1)**  Debris slides with high-mobility causing complete vegetation removal

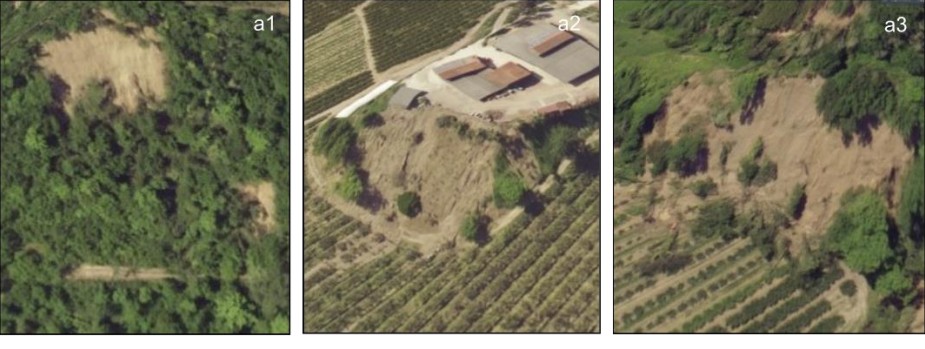

**DS2)** Debris slides with low-mobility causing limited or no vegetation removal

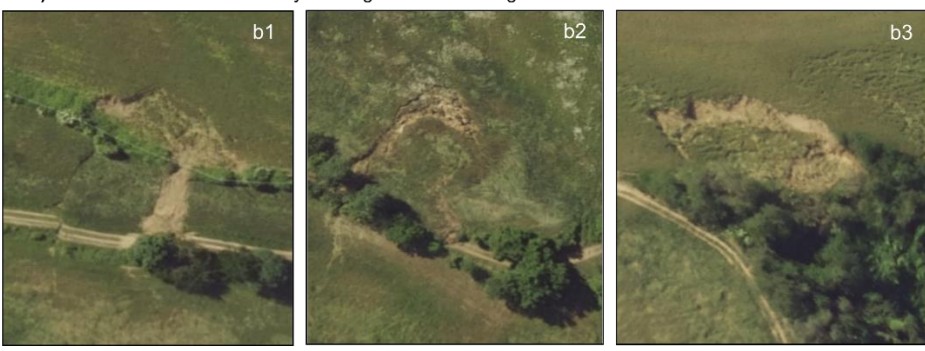

**Fig. 5. Representative images of the two distinct types of debris slides caused by the May 2023 event.**

**RS1)**  Fully-developed rock-block slides

**RS2)** Incipient rock-block slides

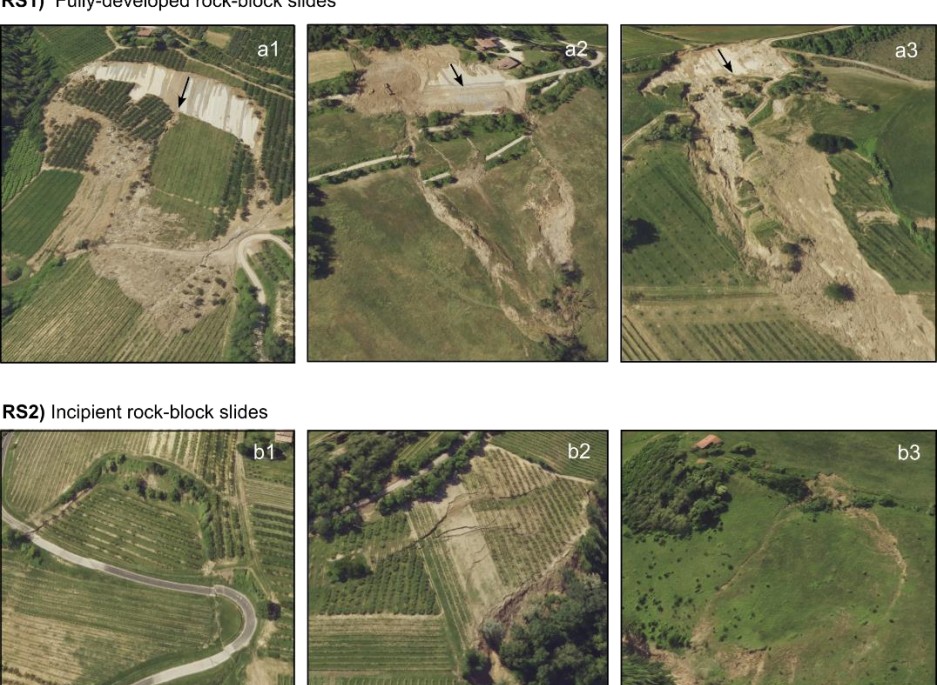

**Fig. 6. Representative images of the two distinct types of rock-block slides caused by the May 2023 event.**

These classification challenges were extensively discussed by our team. We ultimately decided to adhere to the Cruden and Varnes (1996) classification system to define the primary types of landslides. These include debris slides (DS), debris flows (DF), and rock-block slides (RS) in the coarse-grained units (Fig. 1b), along with earth slides (ES) and earth flows (EF) in the fine-grained units. Then, we introduced the informal subclasses of: high-mobility debris slides (DS1), low-mobility debris slides (DS2), long-runout debris flows (DF1), limited-runout debris flows (DF2), fully-developed rock slides (RS1), and incipient rock slides (RS2) to capture the varied behaviors observed in the field. Subclasses were not assigned to earth slides (ES) and earth flows (EF) because landslides in areas with fine-grained soils were significantly less frequent and had milder impacts. This variation in response is likely tied to the distinct hydrological behaviors of fine-grained soils within the study area. Previous studies, which includes statistical analyses of critical rainfall (Rossi et al., 2010; Berti et al., 2012a) and field monitoring of unstable slopes (Berti et al., 2010-2012b), indicates that these clay-rich soils are more prone to failure during extended periods of rainfall rather than brief, intense downpours that generally cause surface runoff and flooding.

**3.4 Quality control**

To ensure the consistency of the results, an experienced geologist conducted a comprehensive review of the entire area after the completion of the manual mapping. This critical review focused on several key aspects to verify that all mappers adopted the same standards of detail and accuracy that were established at the start of the work. Guided by a 1x1 km grid, the reviewer assessed: i) any missed landslides; ii) the precision in outlining landslide boundaries; iii) the consistency in interpreting vegetated areas; iv) the adherence to the established classification criteria; v) the accurate segmentation of individual slide events.

The findings from this review were summarized in a report sent to the twelve mappers. The report ranked the need for revisions in each municipality from "small" to "high" and included a detailed explanation of the necessary adjustments along with screenshots highlighting the errors detected. Each mapper was then tasked to revise their section of the manual inventory based on this feedback. This review and revision phase lasted approximately two months.

Following these adjustments, the landslide map was significantly improved. Although some variations persisted in the resolution of digitization and in interpreting boundaries obscured by vegetation, the primary discrepancies were effectively addressed and resolved. The only remaining issue was the variation in landslide classification among mappers, that was evident when comparing the inventory maps of different municipalities. To address this, we developed the automatic procedure detailed in the next section.

**3.5 Data harmonization**

As mentioned earlier, the criteria for landslide classification were extensively discussed among us. We collectively examined and analyzed approximately 50 complex cases of landslides with uncertain classifications, particularly those that straddle the categories of slides and flows, and incipient rock-block slides. This was done to synchronize the understanding among mappers and promote a consistent approach to classifying these phenomena. These preparatory efforts resulted in a substantial homogeneity in the classification of rock-block slide phenomena. However, notable variations were still evident in the final map between different types of debris slides and flows, which were influenced by the subjectivity of the mappers. Some determined the classification based on the landslide's shape, others on the presence of flow-like features in the deposition area, and yet others on the texture of the debris within the landslide polygon. On the other hand, such distinction is inherently subjective due to the gradual transition between slides and flows, making it

challenging to establish a clear-cut boundary or to define strict classification rules. The same issue arose with classifying landslides based on their degree of mobility. While it was evident that many landslides exhibited complete fluidization and high mobility, significant discrepancies persisted in how the mappers categorized these events.

To address this issue, we implemented the automated procedure depicted in the flow chart of Fig. 7. This procedure utilizes standardized criteria to ensure uniform classification of landslides throughout the area and to correct inevitable errors in such a large dataset. The automated procedure was applied to all landslides except for rock-block slides, which have distinctive features that all mappers clearly and consistently recognized. Four key steps were identified to achieve a consistent classification of material type, movement type, degree of mobilization of debris slides and degree of fluidization of debris flows.

### 3.5.1 Material type (debris or earth)

The initial step was verifying the classification of material types. As previously mentioned, landslides on soil-covered slopes were divided into "debris" and "earth" categories according to the Cruden and Varnes (1996) classification. In our study area, this classification is clearly defined by the underlying bedrock geology; "debris" is derived from coarse-grained rock units, and "earth" from fine-grained units (Fig. 1b).

All mappers employed this classification system, referencing the geological map of the Emilia-Romagna region at a 1:10,000 scale, which provided an objective and standardized framework for classifying material types. However, manual mapping led to inconsistencies due to human error and subjective judgments, particularly when categorizing landslides spanning multiple material types. To address these issues, we overlaid landslide polygons on the lithological map (Fig. 1b) and classified each landslide as either 'debris' or 'earth' based on its polygon's centroid location. This classification was achieved through a simple spatial join between the landslide data and the lithological map within the GIS environment.

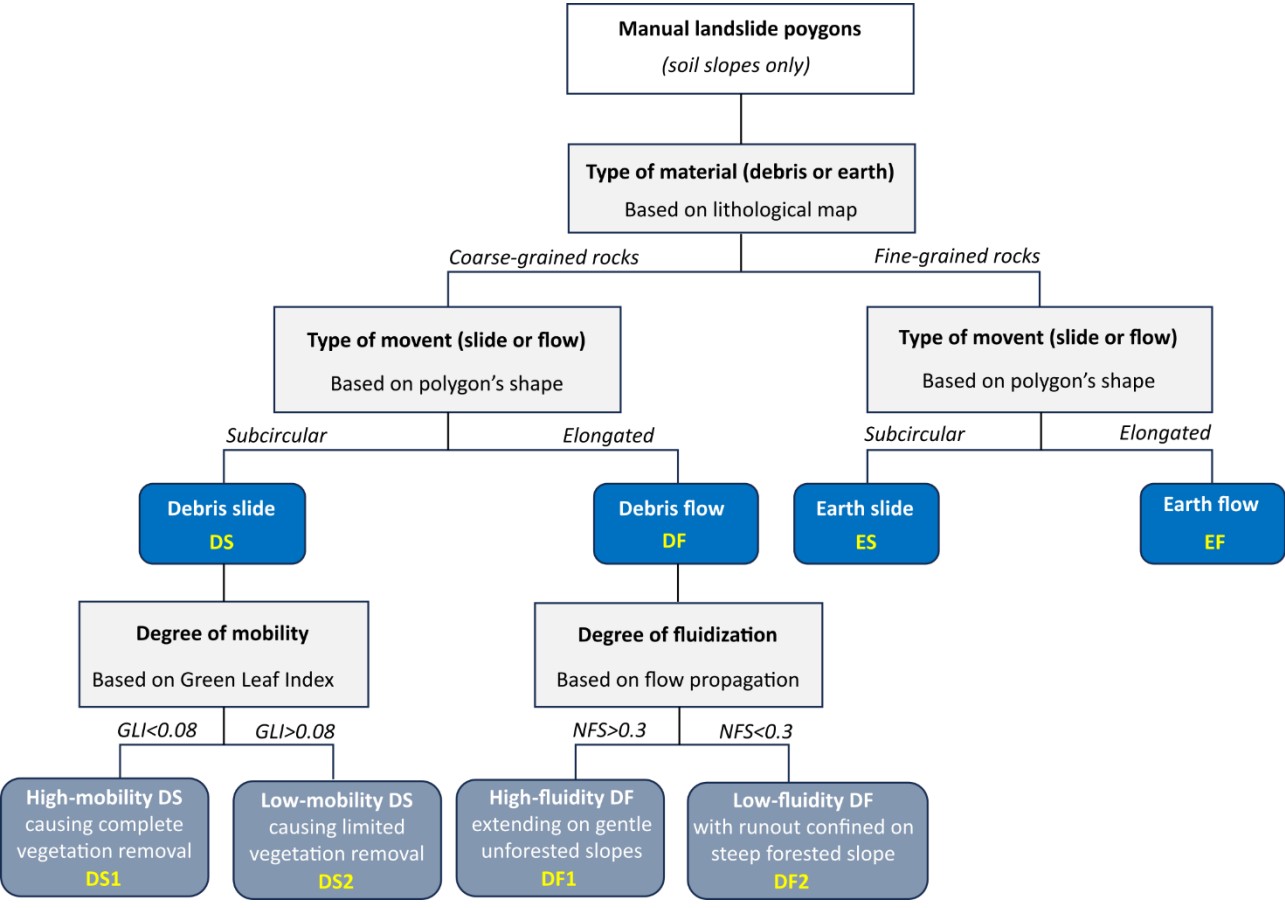

**Fig. 7. Flowchart depicting the process used to ensure data quality and standardize the classification of manually mapped landslides.**

### 3.5.2 Type of movement (slide or flow)

To standardize the distinction between slides and flows, we employed a standard Convolutional Neural Network (CNN) specifically designed to recognize the distinct shapes of slides and flows. The CNN was trained with data from the Casola Valsenio municipality. This area was chosen due to its highly accurate manual mapping and the thorough analyses carried out using multiple mapping techniques (Berti et al., 2024 - " Automated Mapping During an Emergency: Lessons Learned from the 2023 Landslide Event in Romagna, Italy", under review). Moreover, Casola Valsenio served as the initial training ground for the mappers and is the area where classification challenges were collaboratively discussed.

The CNN features an input layer, two convolutional layers (each with batch normalization and ReLU activation), and subsequent max pooling layers to reduce image dimensions and enhance feature extraction. The input layers processes 300x300 pixel black/white images of individual landslides, with landslide areas marked in white. These features are then categorized into 'slide' or 'flow' through a fully connected layer, followed by a softmax layer that determines the class probability. To enhance the model's ability to generalize, we implemented various data augmentation techniques, including random horizontal reflections, rotations ranging from -90 to 90 degrees, scaling from 80% to 120% of the original size, and translations up to 10 pixels.

The network was trained using a randomly selected half of the 4156 debris slides and 1115 debris flows identified in Casola Valsenio, while the other half was utilized to fine-tune the network's hyperparameters and to test and assess the

model's performance. These evaluations showed that the CNN effectively replicates expert classifications of slides and flows. Utilizing the Adam optimizer with an initial learning rate of 0.001 over 100 epochs, the CNN reaches an F1-score

of 0.80 on the testing dataset, indicating robust accuracy in terms of both precision and recall. Fig. 8 displays the confusion matrix obtained for the testing dataset, alongside a selection of landslide images that were correctly and incorrectly classified by the neural network. Of course, as clearly evident looking at the False Positives and False Negatives cases, the CNN cannot overcome the inherent ambiguity in classifying landslides that fall between slides and flows, particularly those that are only partially fluidized and whose polygon shapes are neither distinctly sub-circular nor clearly elongated.

However, by implementing the network across all the polygons, we ensure that the classification criteria agreed in Casola Valsenio are consistently applied throughout the entire area.

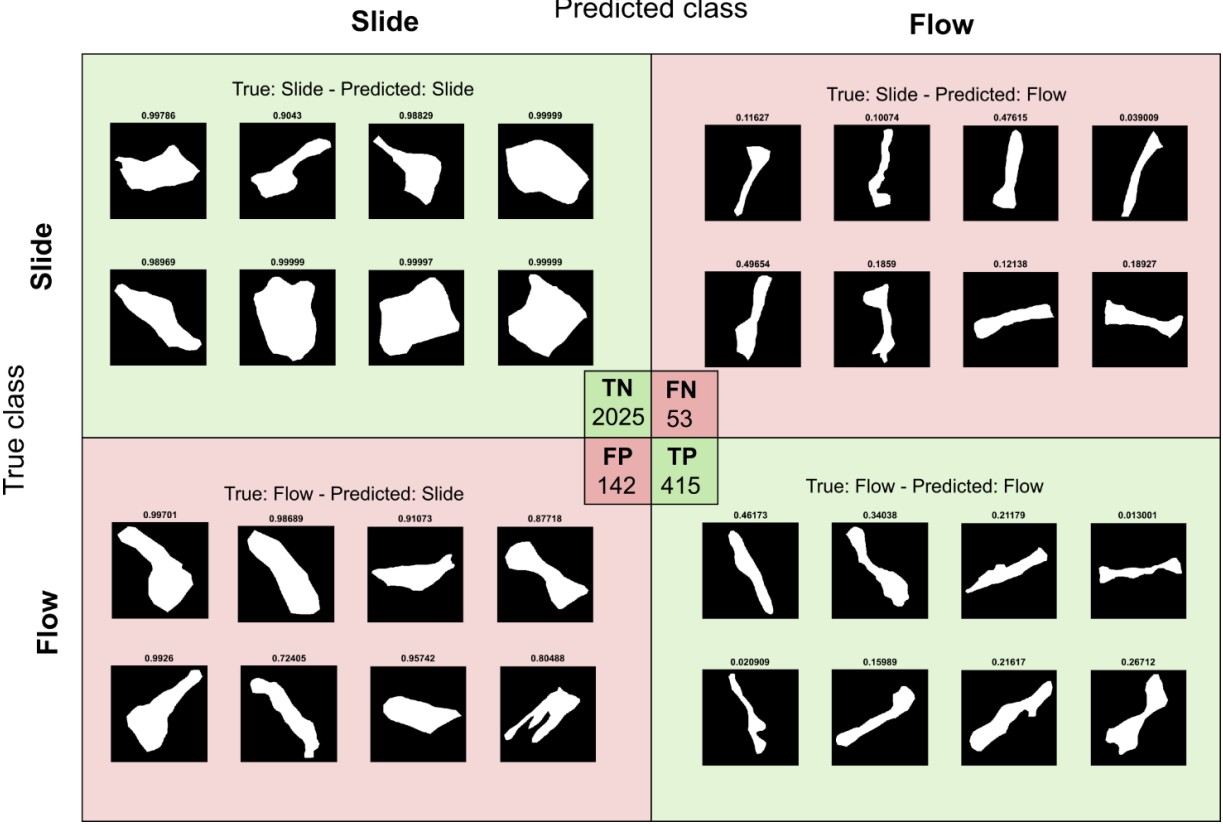

**Fig. 6. Outcomes from the Convolutional Neural Network model applied to differentiate slides from flows using the shape of the polygons. The figure displays the confusion matrix for the testing dataset, which includes 50% of the landslides manually**

**mapped in the Casola Valsenio municipality: TN=True Negative; FN=False Negative; FP=False Positive; TP=True Positive. The small polygons in each category represent example landslides that are correctly (TP, TN) or incorrectly (FP, FN) classified.**

### 3.5.3 Degree of mobility of debris slides

Debris slides were classified by mappers into two categories based on their apparent mobility (Fig. 5). The class DS1, indicating high-mobility slides, was assigned to slides that showed extensive internal disruption and complete removal of

320 vegetation. The class DS2, denoting low-mobility slides, was assigned to slides with minimal internal deformation and little impact on vegetation. While mappers collectively agreed on these criteria for classifying debris slides, discrepancies arose due to variations in personal judgment.

To standardize this assessment, we evaluated the mobility of debris slides by examining the remaining vegetation cover after movement. The Green Leaf Index (GLI) was used to quantify the amount of green vegetation within a landslide polygon:

$$GLI = \frac{(2 \cdot Green - Red - Blue)}{(2 \cdot Green + Red + Blue)} \tag{1}$$

Here, *Green, Red, Blue* denote the reflectance values from the respective color bands. The GLI ranges between -1 and 1, with higher values indicating a denser presence of green leaves.

To identify a suitable GLI threshold for distinguishing the two classes, we analyzed the frequency distribution of GLI values for DS1 and DS2 in the Casola Valsenio dataset. As shown in Figure 9, a distinct separation is observed in the higher categories: 99% of the 4144 high-mobility debris slides (DS1) have GLI values under 0.08, indicating they are primarily bare soils with minimal or no vegetation. In contrast, 34% of the 125 low-mobility slides (DS2) exceed this threshold, suggesting the presence of vegetation. Such a threshold therefore allows for an effective classification of DS1, but it risks misclassification of DS2.

The challenge in distinguishing the two classes stems from the inherent subjectivity involved, especially when vegetation is only partially removed. Like the difficulty in differentiating between slides and flows, no automated method can fully address this issue. However, in our case, the occurrence of DS2 is significantly less frequent than DS1. Consequently, we have chosen to set a GLI threshold of 0.08, acknowledging that this may lead to some misclassification errors with DS2. This approach classifies nearly intact vegetation slides as low-mobility (DS2) and those with partial vegetation as high-mobility (DS1). The resulting F1-score is 0.86, and this threshold has proven stable whether computed on a randomly selected subsample or a specific segment of the Casola Valsenio area.

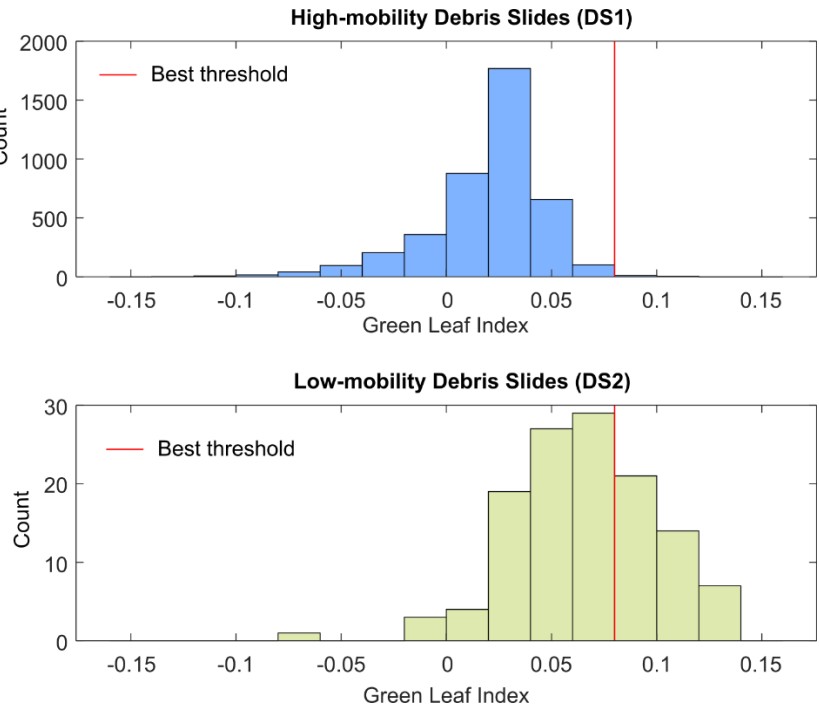

**Fig. 9. Comparison of the frequency distributions of the Green Leaf Index for high-mobility debris slides (upper) and low-mobility debris slides (lower) manually mapped in Casola Valsenio municipality. The red line marks the optimal threshold distinguishing the two landslide types.**

### 3.5.4 Degree of fluidization of debris flows

Debris flows were divided into two distinct classes to highlight differences in fluidization and runout (Fig. 4). The DF1 category was used for long-runout debris flows, marked by fluidized deposits spreading over relatively flat terrain. Conversely, DF2 was used for debris flows with more limited fluidization, typically confined to steep, forested slopes. Mappers used these criteria but also looked at factors such as the size of the debris flow, the presence of a channel, or the location of the initiation area. As a result, the manual classification of DF1 and DF2 was notably inconsistent.

The classification problems are evident when attempting to define an automatic standardization procedure. Both DF1 and DF2 exhibit elongated shapes and absence of vegetation within the landslide areas. Consequently, previous methods that rely on polygon shape or vegetation cover are not applicable. One potential approach could involve using the mean slope of the landslide area, which is generally lower for DF1. However, this metric could introduce bias into the dataset, particularly when comparing the morphological characteristics of the different landslides.

After experimenting with various factors and machine learning techniques, we decided on a simple, reproducible method. Using the Casola Valsenio dataset again, we determined that a reliable indicator of debris mobility is the percentage of the landslide area that extends over Non-Forested Slopes (NFS). NFS encompasses all slopes lacking forest cover, that in most cases are shrub and/or grassy areas, areas with sparse or no vegetation, and agricultural lands. Mappers typically classified debris flows that overrun these areas as DF1. NFS is simply given by:

$$NFS = \frac{A_{NF}}{A_{Tot}} \tag{2}$$

Here $A_{NF}$ is the landslide area on non-forested slopes and $A_{Tot}$ is the total area of the landslide. $A_{NF}$ was detected by overlapping the landslide polygon with the soil use coverage SU2014 provided by the Emilia-Romagna region. This coverage was derived from aerial images captured between May and September 2014, using four bands at a 0.5 m resolution, and classified according to the Corine Land Cover directive. All the slopes not categorized as 311 (Broad-leaved forest), 312 (Coniferous forest), or 313 (Mixed forest) were identified as non-forested.

In Casola Valsenio, a total of 1053 debris flows were documented. Among these, 471 were notably fluid and mobile (DF1), whereas the remaining 582 exhibited less mobility (DF2). The Non-Forested Slopes (NFS) values distinctly varied between the two classes, with DF1 generally displaying higher NFS values (Fig. 10). An NFS threshold of 0.3 has proven to be effective in distinguish between DF1 and DF2: 83% of the DF1 category exceed this threshold, whereas 82% of DF2 falls below it. The corresponding F1-score is 0.82, reflecting a high degree of accuracy.

The harmonization procedure described above resulted in significant modifications to the initial manual classifications. Approximately 50% of the debris slides with limited mobility (DS2) were reclassified as debris slides with high mobility (DS1) due to either heavy or partial clearing of vegetation cover by the movement. About 25% of debris flows (DF1 and DF2) were reclassified as debris slides (DS1 or DS2) due to the limited elongation of the deposit, and about 60% of earth flows (EF) were reclassified as earth slides (ES) for the same reason. It is important to stress that the harmonization process should not be viewed as an automatic classification but rather as an effort to apply consistent classification criteria across the entire area.

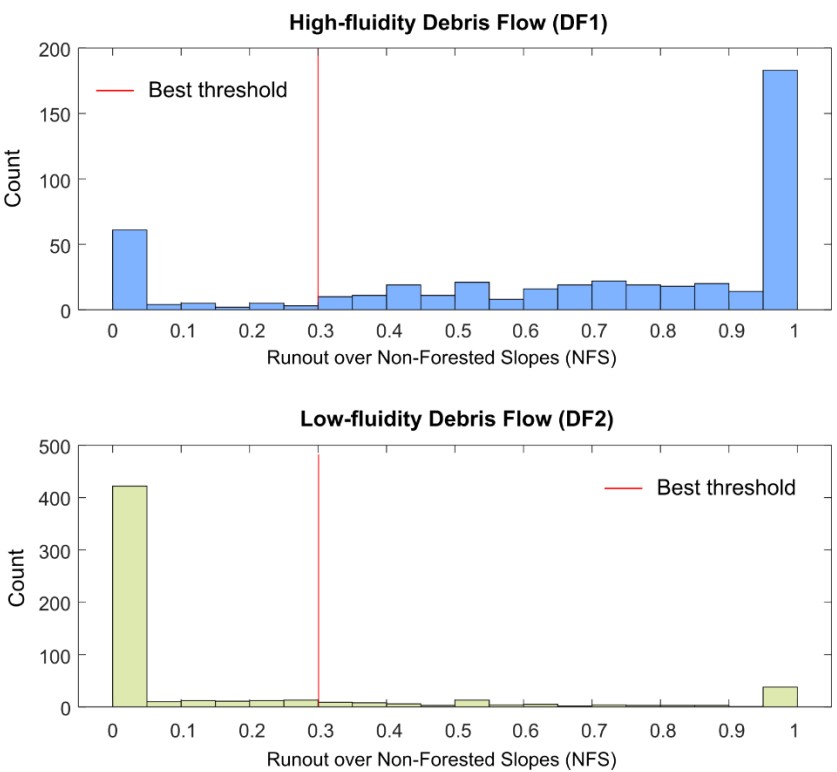

**Fig. 10. Comparison of debris flows with high fluidity (upper) and low fluidity (lower) in the Casola Valsenio municipality, analyzed through the ratio of runout over non-forested slopes (NFS). The red line indicates the optimal threshold for distinguishing between the two types of landslides.**

## 4 The landslide inventory dataset

The landslide inventory for the 2023 Emilia-Romagna event includes 80997 landslide polygons, each categorized according to the classification described in section 3.2. The inventory encompasses landslides triggered by the combined rainfall events of May 1-3 and May 15-16, 2023, without distinguishing between the two events. Differentiation between the events is feasible only in specific small areas where high-resolution images were available after the first rainfall; however, this distinction is not included in the current dataset.

Figure 11 provides an overview of the inventory, showing landslide points (Fig. 11a) and a kernel density map (Fig. 11b). Notably there is a strong, though not perfect, correlation between cumulative rainfall and landslide density. In the eastern part of the region, known as Romagna, the 300 mm rainfall isohyet roughly outlines the area where landslide density exceeds 40 landslides per km². In contrast, the western part of the region, known as Emilia, has a landslide density below 40 landslides per km² despite receiving the same amount of rainfall. This difference can be attributed to the distinct geological settings of the two areas. As shown in Fig. 1b, the Romagna region is primarily characterized by a Miocene flysch (Marnoso-Arenacea Formation, unit 7), which results in steep slopes and coarse-grained weathered soil. Meanwhile, the Emilia region has a more complex geological setting, including extensive areas of fine-grained rocks that responded less intensely to these rainfall events.

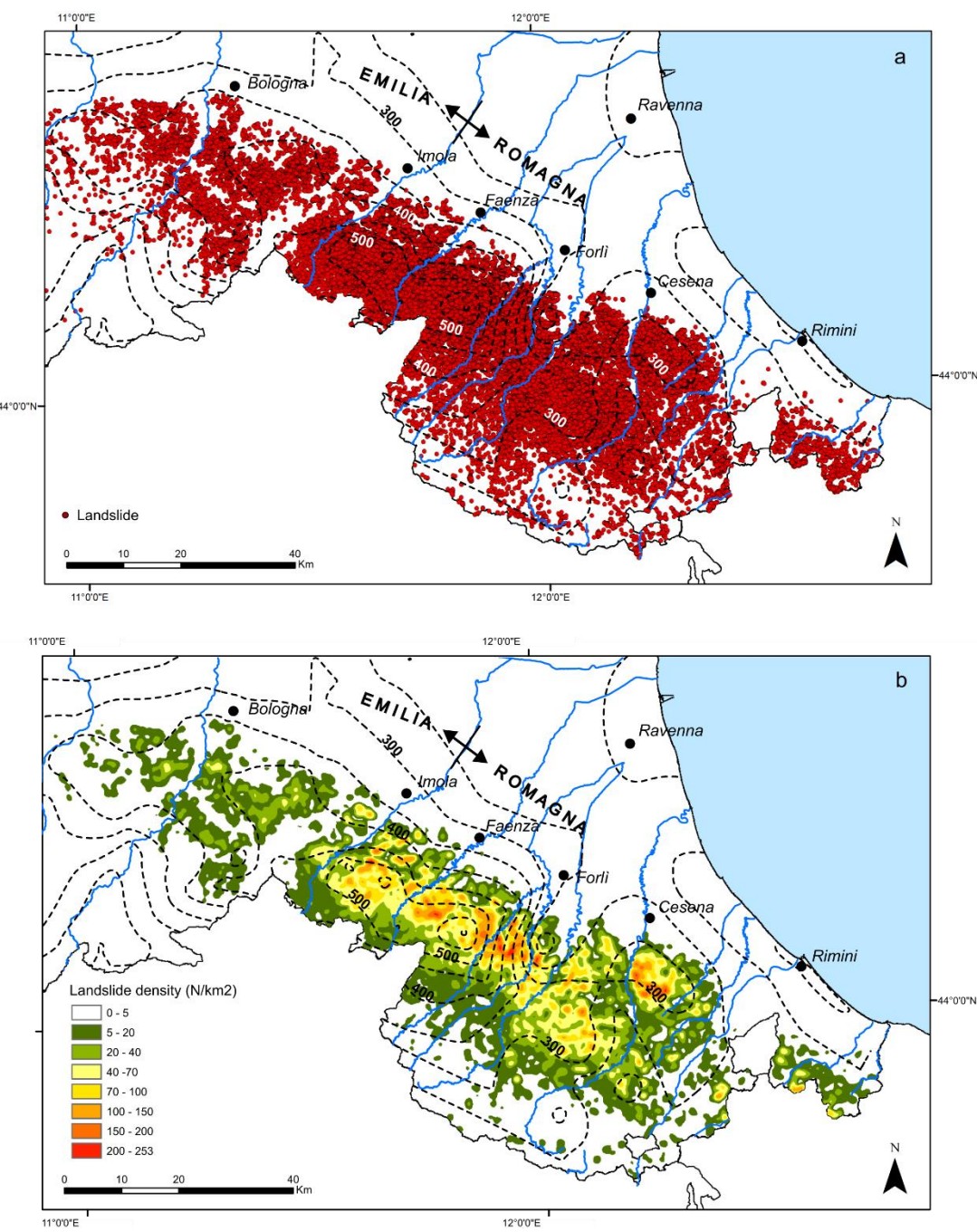

**Fig. 11. a) Map showing the distribution of the 80997 landslides triggered by the May 2023 event in the Emilia-Romagna region, manually mapped and represented as individual points. b) Density map calculated as the number of landslides per 1 km² cell.**

In the Romagna region, landslide density reached an impressive level of over 250 landslides per km². The zone most heavily affected, with more than 40 landslides per km², stretches across roughly 800 km² and covers the outer sector of the Marnoso-Arenacea Formation (red area in Fig. 12). About 64% of the landslides occurred within this zone. The landslide index, which is the ratio of landslide area to total area, reaches in this area the 20-25%. These figures are particularly significant considering they represent the percentage of the area destabilized during a single episode.

A deeper examination of the harmonized inventory underscores the occurrence and main features of various landslide types, providing insights into their spatial distribution and contributing factors. Figures 13 and 14 display several statistical details about the count, dimensions, and slope angles of these landslides. The results from these diagrams are discussed below, enhanced with additional observations from manual mapping and field surveys.

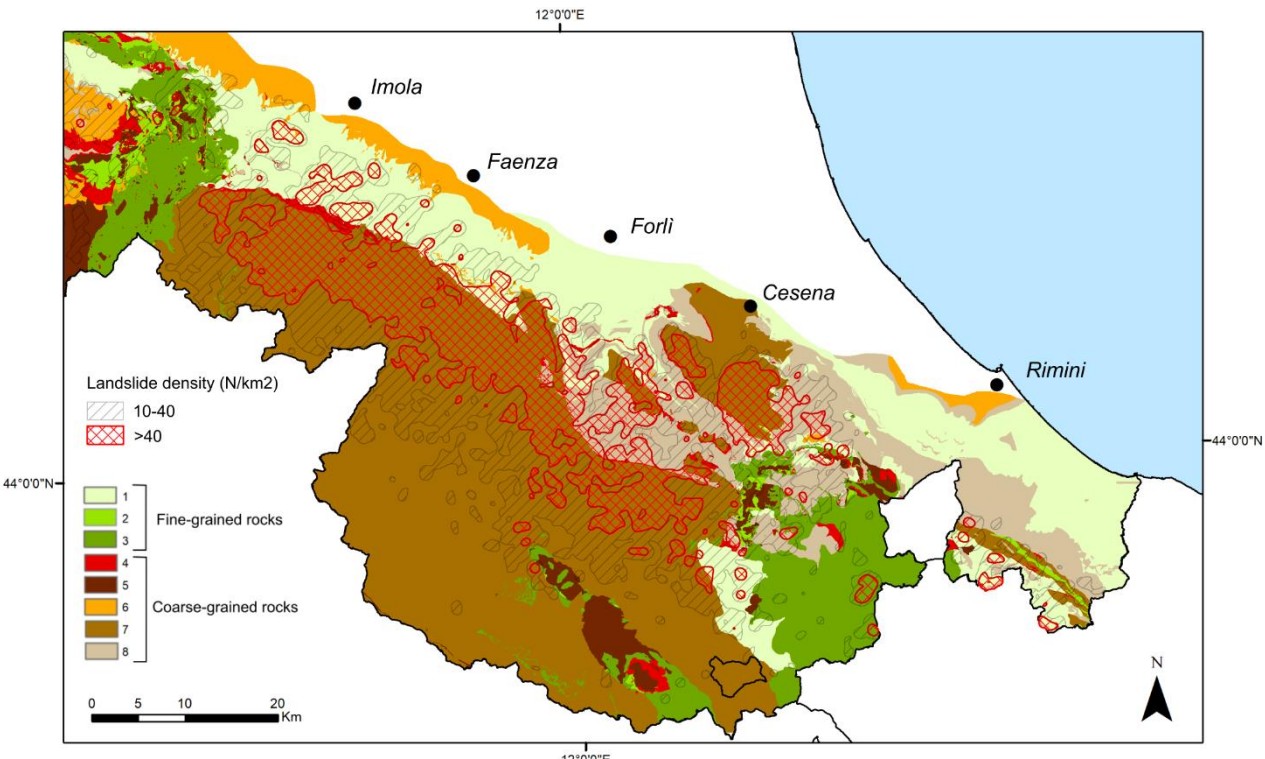

**Fig. 12 Detail of the Romagna area showing two classes of landslide density (between 10 and 40 landslide/km$^2$ and more than 40 landslide/km$^2$) with the eight lithological units in background.**

Debris slides (DS) represent 66% of all landslides by number and 49% by area, marking them as the most common type triggered by the event (Fig. 13). These landslides were generally small to very small (area less than 1000 m$^2$, Fig. 14) and typically occurred on steep slopes with inclines exceeding 25°-30°. In the region, many of these slopes are covered with forests, as they are unsuitable for farming; hence, while root reinforcement and rainwater interception by the tree canopies exist, they were insufficient to prevent these failures. Approximately 94% of the slides were fast-moving and became liquefied after traveling a short distance (DS1). A minor fraction (6%) moved as a coherent mass, showing considerably less internal disruption (DS2). Slides with high mobility caused extensive damage to roads, buildings, and infrastructure and transported large amounts of debris and wood into rivers. Conversely, low-mobility slides predominantly occurred on milder slopes and near roadways. These slides might indicate early-stage slides that had not fully developed, secondary failures behind landslide headscarps, or slides involving rotational movements.

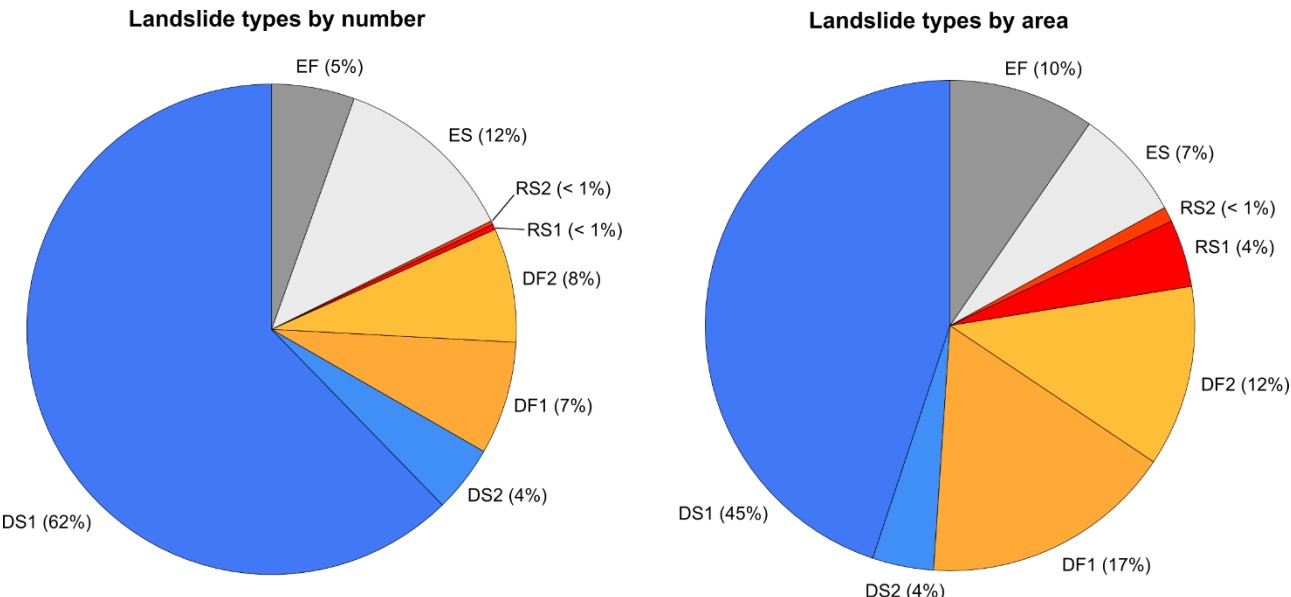

**Fig. 13. Pie charts showing the percentage of landslide types by number (left) and by area (right).**

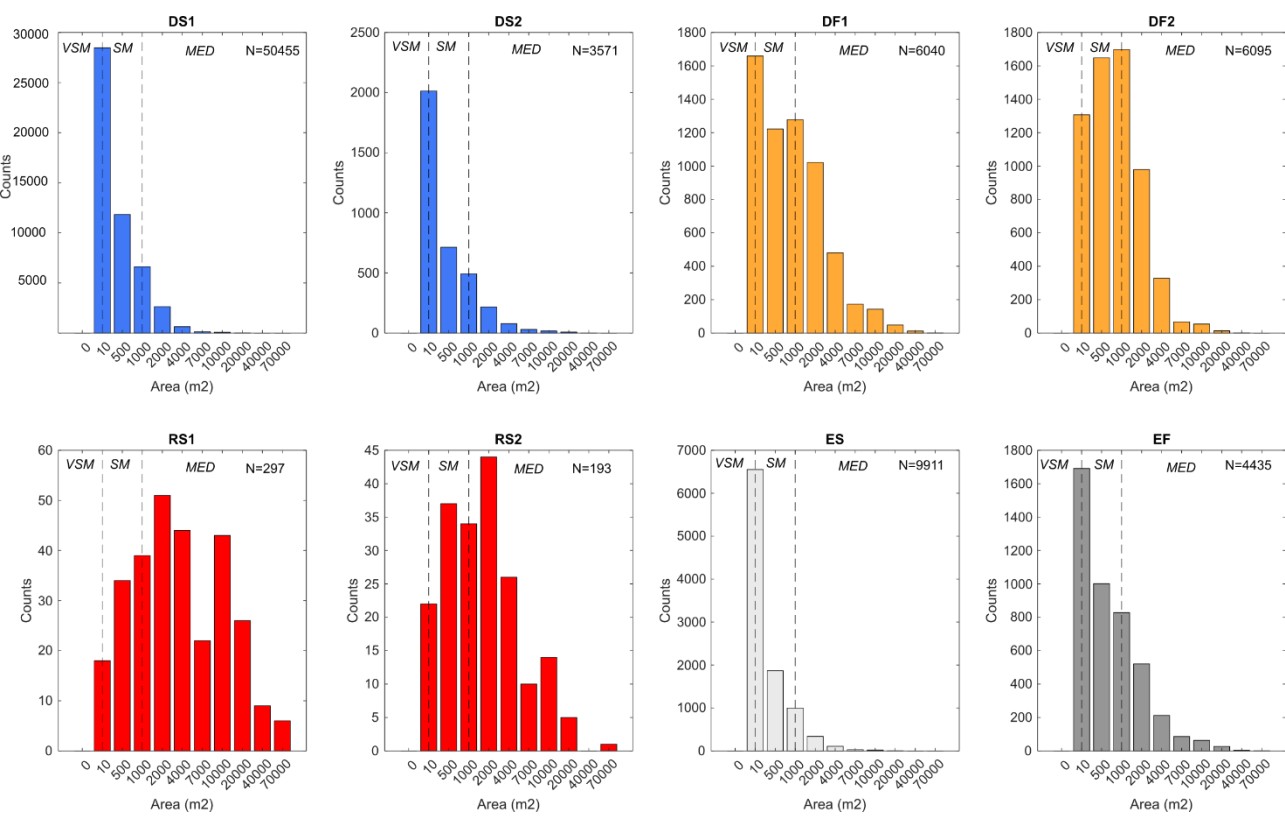

**Fig. 14. Frequency histograms depicting size of various landslide types. The classes labeled at the top of each chart (VSM=very small; SM=small; MED=medium) correspond to the size classification proposed by McColl and Cook (2024).**

Debris flows (DF) are the second most frequent type of landslides, constituting 15% of the total count and 29% by area (Fig. 13). DF were consistently initiated by debris slides on slopes that are generally steeper than 25°-30°. Currently, it remains difficult to ascertain why some debris slides transformed into debris flows while others did not. However, it is

evident that the predominant failure mechanism during the May 2023 event was shallow sliding of the weathered soil cover. Together, debris flows and slides represent 81% of the landslides cataloged in the inventory. After the initial failure, debris flows generally traveled without following predefined channels, and cleared the vegetation forming straight, elongated rectangular shapes. About 50% of these flows had relatively limited runouts halting along steep slopes (DF2), while the rest displayed significant higher mobility, spreading extensively across gentle slopes due to complete fluidization of the material (DF1). A notable feature of the debris flows triggered during the event, especially DF1, was their relatively low destructive power. In many cases, these flows approached buildings and roads without causing substantial damage and spread over grassy fields without harming the vegetation. The limited damage caused by these flows can be linked to their composition, primarily liquefied sand and silt without large cobbles or boulders. This composition enabled them to flow downslope as a dense slurry without a destructive bouldery front. The typical dimensions of debris flows range from "very small" to "medium" (Fig. 14).

Rock-block slides (RS) constitute less than 2% by number and 5% by area of all landslides (Fig. 13), but they left the most profound impression on the public and media. These landslides occurred on homoclinal slopes within the Marnoso-Arenacea Formation (lithological unit 7) and developed as planar slides along bedding planes that aligned with the slope. The thickness of the displaced rock mass varied from about 2 meters to over 30 meters, and several slides extended over areas larger than 10 hectares. Compared to debris flows and debris slides, rock-block slides affected more gentle slopes, typically less than 15° and were bigger in size (class "medium" Fig. 14). Their large volume, high velocity, and the fact that they occurred on sloping lands that were heavily urbanized and farmed made these landslides a major concern during the event. All rock slides initially traveled as coherent rock blocks, moving translationally for several to tens of meters. However, some slides disintegrated during their motion, transforming into rapid flows of debris and fragmented rock. This disintegration typically occurred when the displaced blocks tumbled down an existing scarp or struck a lateral slope, causing the material to break apart. These fragmented rock-block slides were highly mobile and covered long distances.

Earth slides and earth flows accounted for 12% and 5% of the total number of landslides, and 7% and 10% of the total area affected, respectively (Fig. 13). These landslides predominantly occurred within fine-grained units, specifically the Pliocene clays (unit 1) and Cretaceous clay shales (unit 3), as illustrated in Figure 1b. These regions generally experienced fewer landslides, with less severe impacts compared to areas dominated by coarse-grained rocks in the southern parts. The distinct patterns of landslides in areas with coarse- and fine-grained lithological units are clearly illustrated in the sample maps of Figure 15. While landslides are commonly found on steep slopes in both cases, the coarse-grained units also show that even gentle slopes are impacted by extensive long-runout debris flows (DF1 in Fig. 15a) and rock-block slides (RS1). Conversely, in the fine-grained units (Fig. 15b), earth flows and earth slides are primarily concentrated in the badlands areas, with gentle slopes remaining largely unaffected. Moreover, in Emilia-Romagna, it is typical for earth slides and especially for earth flows to occur repeatedly at the same locations as reactivations of dormant landslides. This recurring pattern was evident during the May 2023 event, with most landslides appearing as reactivations of previously known landslides, which were already familiar to the local communities. In contrast, nearly all the landslides in the coarse-grained units—including debris slides, debris flows, and rock-block slides—represented first-time failures and occurred unexpectedly on slopes previously free of documented landslides.

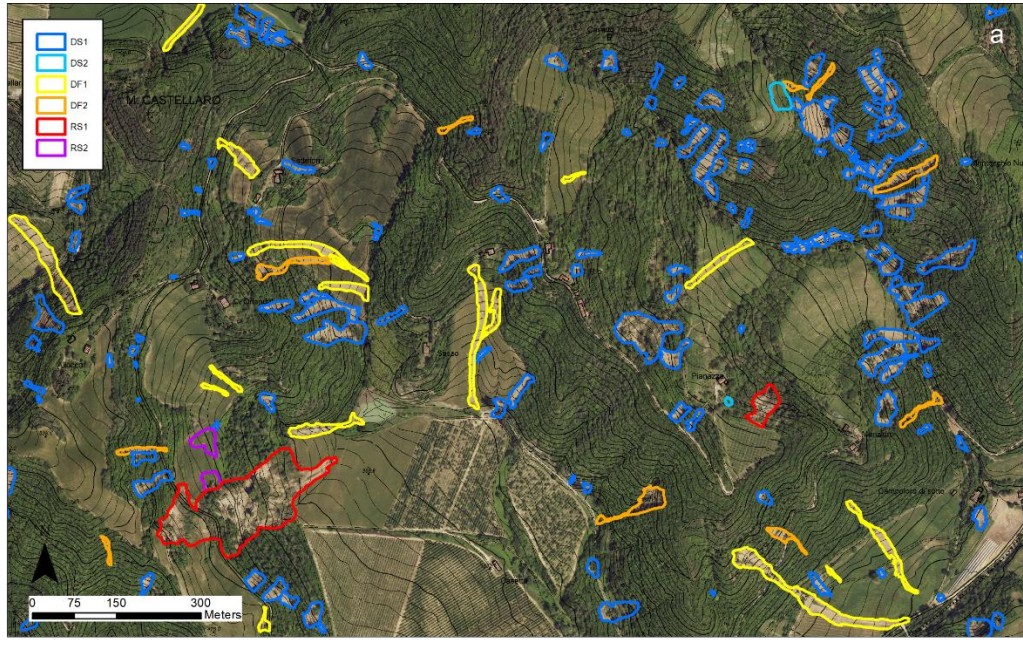

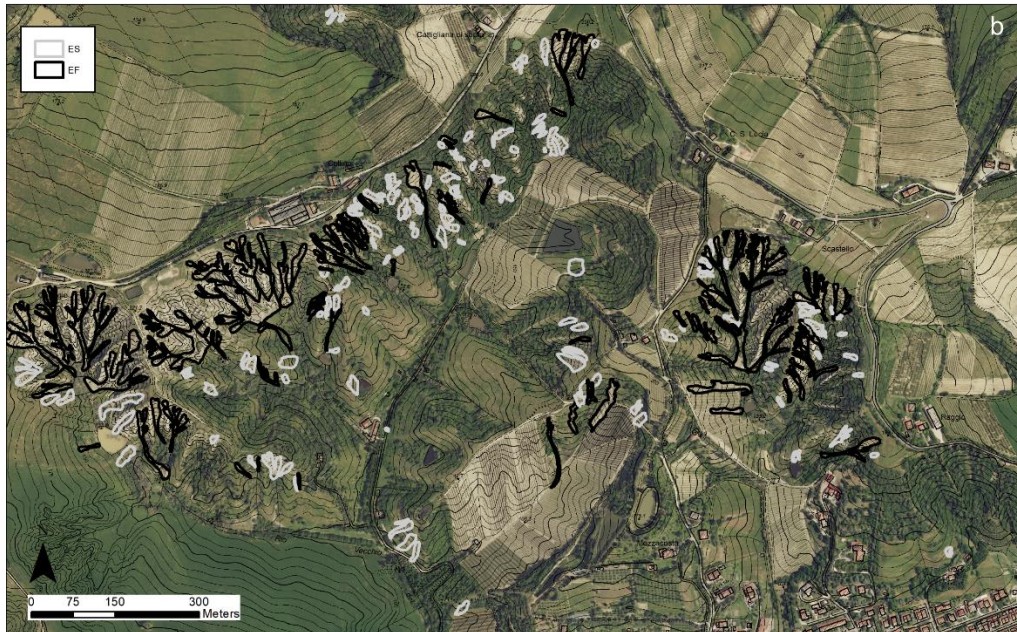

**Fig. 15. Example images from the landslide inventory showing two representative area in coarse-grained units (a) and in fine-grained units (b). DS1=debris slide with high mobility; DS2=debris slide with low mobility; DF1=debris flow with high fluidity; DF2=debris flow with low fluidity; RS1=fully-developed rock-block slide; RS2=incipient rock-block slide; ES=earth slide; EF=earth flow.**

## 5 Limitations, and future updates

The landslide inventory was carried out with great care during the constraints imposed by the emergency situation and ongoing recovery efforts. Utilizing high-resolution 0.2 m imagery and an automated harmonization process has facilitated a detailed and consistent record across the region. However, as is typical with any expert-driven inventory, errors and inconsistencies are unavoidable.

Missed landslides, or false negatives, are likely to occur in shadowed areas like river gorges or steep slopes, as well as in forested areas where landslides have occurred without clearing the vegetation. Additionally, landslides with minimal ground displacement, although clearly visible on-site, may not be discernible in aerial images and thus may remain undetected. False positives—areas mistakenly identified as landslides—are also possible but expected to be fewer. These may include anthropogenic debris accumulations, excavation activities, or plowed fields that alter the soil surface in a manner similar to landslides, or landslides that happened after the pre-event images from April-July 2020 but before the May 2023 event. Nevertheless, we estimate that the combined total of missed or incorrectly identified landslides might constitute less than 1% of the total inventory.

The primary limitation of the inventory likely lies in the accuracy of the landslide boundaries. Not all mappers across the area had sufficient time to delineate the landslide polygons with high-resolution detail, resulting in some boundaries appearing jagged and imprecise upon closer inspection. The data quality procedures brought this issue to light, but redrawing all the rough-edged polygons would be excessively time-consuming. Given that the locations of these landslides are accurate, we chose to publish and make available the inventory in its current state and defer any refinements to future updated versions that, besides refining polygons, will also incorporate changes recommended by local authorities, which are currently underway. In April 2024, the Emilia-Romagna region shared the landslide inventory with all municipalities affected by the event, requesting feedback on any overlooked landslides. This information is now being gathered, and a first update is scheduled for completion by the end of 2024. Initial feedback primarily concerns small landslides that caused damage to private or public properties but were not detected in aerial photographs due to minimal displacement. These new landslides will be included in Version 2 of the inventory, which will be available in the same Zenodo repository.

**6 Data availability**

The landslide inventory is freely accessible in the Zenodo repository (DOI: 10.5281/zenodo.13742643, Pizziolo et al., 2024). The dataset is available as an ESRI (Environmental Systems Research Institute) shapefile and is compatible with GIS software. The shapefile encompasses several attributes: polygon ID (IDC), landslide type as manually classified by the operator (ClassMan), geological unit of the polygon's centroid (Lito), Green Leaf Index (GLI), percentage of deposit over Non-Forested Slopes (NFS), landslide type after applying the harmonization algorithm (ClassNew).

**7 Concluding remarks**

The dataset of the May 2023 Emilia-Romagna event encompasses more than 80.000 rainfall-induced landslides (mostly first-failure) distributed over an area of more 6000 km², with density reaching as high as 200 landslides/km2. Despite some inherent limitations and potential areas for improvement of the dataset, we believe that our landslide inventory offers significant value to the scientific community and to the involved institutions for several reasons.

Firstly, it documents the response of a large area to an exceptional meteorological event, likely linked to ongoing climate change. This can support the scientific community in proving that multiple occurrences of rainfall-related landslides are likely to become more frequent in the coming years, and can make decision makers more aware of the fact that even slopes that have been unaffected by landslides in the past, cannot be considered free of risk for the future.

Secondly, the Emilia-Romagna relatively straightforward geological framework makes it ideal for conducting geospatial analyses of landslide susceptibility, and to prove that they can be adopted to support land-use planning in addition to landslides inventories. Actually, the Emilia-Romagna region's geoportal provides free access to an extensive range of

spatial data, including DEMs, lithology, land use, and rainfall data, all of which can be integrated with our landslide map to test both traditional and machine-learning-based predictive tools).

Thirdly, the predominance of shallow planar failures in this event provides an excellent case for testing physically-based slope stability models, and to highlight the relevance of such type of landslides in the study area, so to promote a much more careful evaluation of the possible impact of such phenomena on existing infrastructures network and for designing new assets.

In conclusion, we warmly invite interested colleagues to contact us with any questions, specific needs, or to initiate a collaborative research effort that could transform a tragic event into an opportunity to enhance our understanding of landslide risk assessment.

**Author contribution**

Investigation (landslide mapping): Matteo Berti, Michele Scaroni, Mauro Generali, Vincenzo Critelli, Marco Mulas, Melissa Tondo, Francesco Lelli, Cecilia Fabbiani, Francesco Ronchetti, Giuseppe Ciccarese, Nicola Dal Seno, Elena Ioriatti, Rodolfo Rani, Alessandro Zuccarini, Marco Pizziolo, Alessandro Corsini. Conceptualization, Formal analysis, Visualization, Writing (original draft preparation): Matteo Berti. Validation, Writing (review & editing): Alessandro Corsini. Funding acquisition: Tommaso Simonelli. Project administration, Resources: Marco Pizziolo.

**Competing interests**

The authors declare that they have no conflict of interest.

**Acknowledgements**

This study was carried out within the RETURN Extended Partnership and received funding from the European Union Next-GenerationEU (National Recovery and Resilience Plan – NRRP, Mission 4, Component 2, Investment 1.3 – D.D. 1243 2/8/2022, PE0000005).

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
