# Peer review of "RER2023: the landslide inventory dataset of the May 2023 Emilia-Romagna meteorological event"

_Earth System Science Data, 2024_

## Author Comment (AC1)

Dear Reviewers,

Thank you for taking the time to review our manuscript and for your insightful comments. In this response, we address your remarks and detail the changes made to the original manuscript in accordance with your suggestions.

To enhance clarity, we have employed the following text styles:

| | |
|---|---|
| *black, italics*: | your comment |
| blue, plain text: | our reply |
| *blue, italics:* | revised text |

**Reply to RC1**

*The study entitled "**RER2023: the landslide inventory dataset of the May 2023 Emilia-Romagna event**", by Matteo Berti et al. is a laborious study presenting the process and results of the landslide inventory conducted following an extreme rainfall event occurred in May 2023 in Emilia-Romangna region in Italy.*

*Overall, I consider that the paper is well structured, with a large development of the section presenting the research context, with a clear and detailed emphasis on the methodological working steps followed in landslide mapping and with valuable results of landslide typology and their spatial distribution mapped.*

*Although I don't have any doubts regarding the correctness of the methodology applied to map the landslides in the study area, I have a concern regarding the way in which the authors mapped the landslides.*

*During the mapping process, how the authors distinguished between the landslides (re)activated during previous extreme rainfall events, and those landslides resulted during and/or after the 2023 event?*

Yes, we differentiated between existing and new landslides, mapping only those landslides (whether first-time occurrences or reactivations) that took place during the May 2023 event. Indeed, this is a crucial point that we did not emphasize sufficiently. The distinction was clear-cut, as described in the revised text:

*"Overall, the process of identifying landslides was fairly objective. Each landslide cleared vegetation, uncovered distinct patches of bare soil or bedrock, and led to deposits of loose material. The post-event images, taken only 10 days after the second rainfall, clearly displayed these geomorphological markers, eliminating any ambiguity in recognizing the landslides from the event. Additionally, most of the landslides were new occurrences, not present in the pre-event images; even in case of reactivations, it was straightforward to identify the newly affected areas. This gives us confidence that our dataset exclusively contains landslides from the May 2023 event."*

*If an inventory landslide map does exist in the study area with landslide types and their distribution before the 2023 event, can the authors confirm that the same criteria were previously applied to distinguish between various type of landslides?*

Thank you for highlighting this issue. We did not apply the same criteria used by other geologists to map the landslides, as ours were specifically tailored for this particular event. We have clarified this in the revised text:

*"It is important to emphasize the distinct nature of this dataset. The existing landslide map of Emilia-Romagna Region (https://geoportale.regione.emilia-romagna.it/) encompasses all landslides identified in the region via photo interpretation, historical data, and field surveys. This map provides a comprehensive overview of landslide activity across a broad geological timeframe, accounting for various climatic, seismic, and morphological conditions. In contrast, our dataset specifically captures the landslide response to a single critical meteorological event. Additionally, the criteria used to distinguish different types of landslides are specifically designed for this scenario, as explained in the subsequent section (see 3.3)."*

*I have few specific comments, which are listed below:*

*Lines 109 – 110: It is stated here that the existing geological map was used to delineate geological formations. You should indicate the scale of the geological map used for this purpose, in order to allow the reader to appreciate how detailed is the spatial distribution of the lithological units on the map.*

Apologies for the oversight. The geological map is at a scale of 1:10,000. We have now included this information in the text.

*Lines 196 – 198: You stated here that "…fine-grained soils have lower permeability and are more likely to fail during extended periods of rainfall rather than during brief events, where surface runoff and flooding are more prevalent". How do you know that, is this from your previous observations, or this is coming from other sources (in this case, you should indicate the references)? Please explain this.*

This sentence reflects an insight we gained from our research in the area. Admittedly, it was not sufficiently substantiated. We have incorporated several references in the revised text to provide a more robust foundation:

*"This variation in response is likely tied to the distinct hydrological behaviors of fine-grained soils within the study area. Previous studies, which includes statistical analyses of critical rainfall (Rossi et al., 2010; Berti et al., 2012a) and field monitoring of unstable slopes (Berti et al., 2010-2012b), indicates that these clay-rich soils are more prone to failure during extended periods of rainfall rather than brief, intense downpours that generally cause surface runoff and flooding."*

*Figure 13 (a) and (b): You should add in the figure caption the meaning of the symbol letters attributed to each of the coloured polygons on the map.*

Thank you for your suggestion. We have added the definitions of the symbol letters to the figure caption.

*This is a review of the manuscript titled as "RER2023: the landslide inventory dataset of the May 2023 Emilia-Romagna event" by Berti et al. The main objective of the work is to present the inventory the authors have produced post-2023 rainfall-induced landslide event in the Emilia-Romagna region of Italy.*

*The article is well-written and organized. The overall quality of figures, tables, and equations are good. The use of English language is generally good. The methodology is well described and the outcome, i.e., the landslide inventory, is a valuable resource for further research and application. This reviewer also appreciates the authors' effort to clearly outline the limitations of the work.*

*Even though the paper goes into great details about the background of the research and different steps followed to develop this inventory, a couple of segments could be improved.*

*For example, the segment "Landslide identification and mapping" lacks in specific example of how the identification was done. It could be helpful if the authors could include a case showing the satellite image and how the landslide was identified and the boundary was marked. Adding a validation example will also help the reader on the effectiveness of the method and the uncertainties around the drawn boundary.*

Thank you for your suggestion. In the revised manuscript, we have included two additional figures and expanded the text to enhance the clarity regarding how landslides were identified and the associated uncertainties. Figure 2 illustrates an example of landslide identification, detailing the procedure followed: from pre-event images through post-event RGB, NIR, and NDVI images, to 3D visualization and final mapping. Figure 3 presents examples of uncertain delineation of landslides, specifically in cases of debris slides spreading through vegetation and multiple coalescent slides. The corresponding sections of the text have been updated accordingly:

*"Once a landslide was spotted, further inspection was conducted using NIR and NDVI images (Fig. 2C and D) and supplemented by a 3D viewer featuring high-resolution images overlaid on a 30 m DEM (Copernicus GLO-30, ESA, 2024; Fig. 2E). Viewing the slope from different angles enhanced the delineation of the affected area and the interpretation of the type of movement. Following this analysis, each landslide was classified into the specified classes described in the next section. The digital mapping of the landslide polygon was then executed at scales ranging from 1:800 to 1:200, depending on the landslide's size, ensuring precise tracing of the affected perimeter. In this final stage, the regional topographic map at 1:10,000 scale was utilized to further verify alignment with the existing topography (Fig. 2F)."*

*"While identifying the landslides was straightforward, defining their exact boundaries was more subjective. Many landslides became fluidized upon failure, with the distal debris spreading among trees without removing vegetation, thus complicating the mapping of the deposit (Fig. 3a). Moreover, several slopes experienced complete removal of soil cover by adjoining shallow failures, blurring the distinction between individual slides (Fig. 3b)."*

*The authors could also include an explanation on how they distinguish the landslides triggered by the 2023 event from the existing/previous ones.*

We have responded to this comment, which was also noted by the first reviewer, by clarifying why we are confident that the dataset exclusively includes landslides triggered during the May 2023 events. Here is the text added to section 3.2, 'Landslide Identification and Mapping':

*"Overall, the process of identifying landslides was fairly objective. Each landslide cleared vegetation, uncovered distinct patches of bare soil or bedrock, and led to deposits of loose material. The post-event images, taken only 10 days after the second rainfall, clearly displayed these geomorphological markers, eliminating any ambiguity in recognizing the landslides from the event. Additionally, most of the landslides were new occurrences, not present in the pre-event images; even in case of reactivations, it was straightforward to identify the newly affected areas. This gives us confidence that our dataset exclusively contains landslides from the May 2023 event. It is important to emphasize the distinct nature of this dataset. The existing landslide map of Emilia-Romagna Region (https://geoportale.regione.emilia-romagna.it/) encompasses all landslides identified in the region via photo interpretation, historical data, and field surveys. This map provides a comprehensive overview of landslide activity across a broad geological timeframe, accounting for various climatic, seismic, and morphological conditions. In contrast, our dataset specifically captures the landslide response to a single critical meteorological event. Additionally, the criteria used to distinguish different types of landslides are specifically designed for this scenario, as explained in the subsequent section (see 3.3)."*

*In the segment "Landslides classification", the authors cite several existing methods that have been deployed in the current study. It presumes that readers may already know different landslide classifications. It would be helpful if the authors include a small background on different classification schemes of the cited articles and the one adopted by the current study.*

Thank you for highlighting this issue. Indeed, the original text did not clearly convey the classification system we employed. In the revised manuscript, we have included the following sentence at the beginning of section 3.3 'Landslide Classification':

*"Right from the start of our work, the classification of landslides triggered by the event was identified as a crucial task. We utilized the well-known Cruden and Varnes (1996) classification system, which categorizes landslides based on two primary criteria: the type of movement and the type of material. Most of the landslides from the May 2023 event were categorized either as debris slides, where mixed granular material moves along a plane of weakness as a relatively coherent mass, or as debris flows, where the material moves in a fluid-like manner over greater distances. However, we encountered challenges in further distinguishing between various types of debris slides and debris flows, a differentiation not addressed by the standard classification."*

*Editorial comments:*

*Line 39: "Therefore it is important ……" could be simplified*

*Line 202: "1 km x 1 km" instead of "1 x 1 km"*

*Line 324: ATot in equation 2 is probably not defined.*

Corrected, thank you.

Dear Reviewer, we have addressed all the modifications noted in the PDF. Below, you will find a more detailed explanation of the key points.

*row 52-54: However, these datasets did not aim to, and did not reach, the level of completeness, consistency and accuracy required for updating the official landslide inventory that, in turn, determines significant practical consequences for the management of the mapped areas and their surroundings.*

*RC3 comment: Explain why*

The primary reasons the datasets from Ferrario and Livio (2024) and Notti et al. (2024) are deemed inadequate for updating the landslide inventory of the Emilia-Romagna region include: the limited resolution of the satellite images employed by the authors and the lack of landslide classification. We have now addressed these issues in the revised text:

*"Ferrario and Livio (2023) provided an initial screening of these landslides through visual inspection of Planet satellite images (3 m resolution) and limited field surveys, while Notti et al. (2024) utilized an unsupervised identification method on Sentinel 2 images (10 m resolution). Both inventories were quickly compiled soon after the emergency, utilizing satellite imagery with resolutions that are relatively low considering the small size of the landslides from May 2023. Indeed, many of these landslides were so small that they were challenging to detect even at a 3 m resolution. Additionally, landslides were not classified or categorized by type of movement or material. Consequently, these two datasets were not intended to achieve, nor did they achieve, the level of completeness, consistency, and accuracy required for updating the official landslide inventory."*

*row 109: The he geological map of the Emilia-Romagna region*

*RC3 comment: please add the scale map a reference.*

Scale map and reference have been added:

*"The geological map of the Emilia-Romagna region, created by the regional Geological Survey at a 1:10.000 scale (AGSS-RER, 1986), includes more than 600 geological formations."*

*row 142-144: The digital mapping of the landslide polygon was then executed at scales ranging from 1:800 to 1:200, depending on the landslide's size, ensuring precise tracing of the affected perimeter. RC3 comment: If I understood correctly you have identified and classified the landslides at 1:1000 scale (the screen zoom) but you digitized at a larger scale (1:200 to 1:800). Why did you not digitize at the same scale? Could you indicate what is the suitable consulting scale of the final inventory?*

Sorry for the rough description of the digitalization process. We used (approximately) a 1:1000 scale to identify the landslide at the slope scale, then we employed a large scale (up to 1:200) to precisely draw the landslide boundary following the perimeter of the unvegetated areas or the limit of the

landslide deposit. Thank you for pointing out the issue of the consulting scale of the inventory. It is about 1:2000, and it has been added to the revised text in section 3.2 'Landslide identification and mapping':

*"While the delineation of the landslides was carried out at a large scale for precise mapping of the boundaries, the final inventory is designed to be appropriate for consultation at a scale of 1:2000."*

*Row 144: Identifying the landslides was relatively straightforward and objective*

*RC3 comment: explain why, please.*

This text has been added to better explain why landslide mapping was straightforward:

*"Overall, the process of identifying landslides was fairly objective. Each landslide cleared vegetation, uncovered distinct patches of bare soil or bedrock, and led to deposits of loose material. The post-event images, taken only 10 days after the second rainfall, clearly displayed these geomorphological markers, eliminating any ambiguity in recognizing the landslides from the event. Additionally, most of the landslides were new occurrences, not present in the pre-event images; even in case of reactivations, it was straightforward to identify the newly affected areas."*

*row 147-152: . Moreover, several slopes experienced complete removal of soil cover*

*RC3 comment: could you show examples in a figure?*

We have included two new figures in the manuscript: Figure 2, which illustrates an example of landslide identification, and Figure 3, which displays examples of challenging landslide delineation. The accompanying text has been updated to reflect these additions, aiming to clarify the process of landslide mapping more effectively.

row 215-217: A substantial number of cases were collectively examined and analyzed to synchronize the mappers' perceptions and foster a unified approach to class attribution.

*RC3 comment: what does substantial mean? quantity? different types of example?...please clarify*

We apologize for the lack of precision. The sentence has been expanded as follows:

*"As mentioned earlier, the criteria for landslide classification were extensively discussed among us. We collectively examined and analyzed approximately 50 complex cases of landslides with uncertain classifications, particularly those that straddle the categories of slides and flows, and incipient rock-block slides. This was done to synchronize the understanding among mappers and promote a consistent approach to classifying these phenomena."*

*Usually before starting to map landslides using photo-interpretation, the interpreters discuss, decide and adopt common criteria aimed to investigate the landslides and the structure and rules for the geodatabase (i.e shape file...). The role of the expert geologist is verify  if the mappers applied consistently the common rules and criteria.*

This describes our exact process. Initially, we reviewed the classification criteria, work scale, and the utilization of pre- and post-event images, among other aspects. Throughout the project, we shared and resolved doubts and challenges. Despite these concerted efforts, discrepancies in landslide classification among the mappers became evident. As a result, we implemented a series of automatic classification methods (not automated mapping) to ensure uniformity in the finale map. The entire workflow should be clear in the revised text.

*Fig.13: please modify the legend to refer to the colored polygons. Please move the north arrow from here to the left down corner above the scale bar*

Figure 13, now updated to Figure 15, has been revised to display only the landslide types present in the map within the legend. The north arrow has been placed to the left corner.

---

## Author Response (AR2)

Dear Reviewers,

Thank you for taking the time to review our manuscript and for your insightful comments. In this response, we address your remarks and detail the changes made to the original manuscript in accordance with your suggestions.

To enhance clarity, we have employed the following text styles:

| | |
|---|---|
| *black, italics*: | your comment |
| blue, plain text: | our reply |
| *blue, italics:* | revised text |

**Reply to Natascha Töpfer**

*1. Your reference list includes works "under review". Such works can be cited upon submission if being available to the reviewers. They should not be cited in the final, accepted manuscript, unless published, accepted for publication, or available as preprint with a DOI. 2.*

Okay, as the paper is still under review, we have omitted the reference from the revised text:

*"3.5.2 Type of movement (slide or flow)*

*To standardize the distinction between slides and flows, we employed a standard Convolutional Neural Network (CNN) specifically designed to recognize the distinct shapes of slides and flows. The CNN was trained with data from the Casola Valsenio municipality. This area was chosen due to its highly accurate manual mapping, and because it served as the initial training ground for the mappers where classification challenges were collaboratively discussed."*

*2. Table 1 contains coloured cells or/and coloured values. Please note that this will not be possible in the final revised version of the paper due to HTML conversion of the paper. When revising the final version, you can use footnotes or italic/bold font.*

We have removed the colored cells from Table 1 as they are unnecessary; the different lithological groups (1-3 and 4-8) are explained in the caption.